# Fathers' needs in a surgical neonatal intensive care unit: Assuring the *other* parent

Priya Govindaswamy[1,2]*, Sharon M. Laing[3], Donna Waters[2,4], Karen Walker[2,4], Kaye Spence[1], Nadia Badawi[1,2]

**1** Grace Centre for Newborn Care, Children's Hospital at Westmead, Westmead, Australia, **2** Sydney Medical School, University of Sydney, Sydney, Australia, **3** Western Sydney University, Sydney, Australia, **4** Sydney Nursing School, University of Sydney, Sydney, Australia

* Priya.govindaswamy@health.nsw.gov.au

**Data Availability Statement:** There are ethical and legal restrictions on sharing de-identified data set. Data from the study are available upon request, as there are legal restrictions on sharing these data

## Abstract

### Objectives

Fathers of infants admitted to Neonatal Intensive Care Unit (NICU) play an important role and have individual needs that are often not recognised. While there is considerable evidence regarding mothers' needs in the NICU, information about fathers' is particularly limited. This study identifies the needs of fathers of newborns admitted to NICU for general surgery of major congenital anomalies, and whether health-care professionals meet these needs.

### Methods

Forty-eight fathers of infants admitted for surgery between February 2014 and September 2015 were enrolled in a prospective cohort study. Fathers completed the Neonatal Family Needs Inventory comprising 56 items in 5 subscales (Support, Comfort, Information, Proximity, Assurance) at admission and discharge and whether these needs were met; as well as the Social Desirability Scale.

### Results

Responses showed Assurance was the most important subscale (M 3.8, SD .26). Having questions answered honestly (M 3.9, SD .20) and knowing staff provide comfort to their infant (M 3.94, SD .24) were fathers' most important needs. By discharge, fathers expressed a greater importance on being recognised and more involved in their infant's care. More than 91% indicated their ten most important needs were met by the NICU health-care professionals, with no significant changes at discharge. Clergical visits (M 2.08, SD 1.21) were least important.

### Conclusions

Reassurance is a priority for fathers of neonates in a surgical NICU, particularly regarding infant pain management and comfort. It is important that health-care professionals provide

publicly as these data contains sensitive and identifiable information. The data set contains details including birthweight, gestational age, gender, antenatal diagnosis, surgical diagnosis, fathers' age, and birth country. Such information may be used to directly identify individuals, as the study site is one of only 3 state-wide referral centers in New South Wales for neonatal surgery of major congenital anomalies and each year there are relatively few neonates requiring surgery for each specific surgical diagnoses. The informed consent signed by the study participants and approved by the Internal Ethics Review Committee of the Children's Hospital at Westmead, did not ask fathers about data sharing. Researchers who meet the criteria for access to confidential data may contact the corresponding author or the Executive officer, KIDS Research, Asra Gholami, 02-98453066 or asra.gholami@health.nsw.gov.au, and provide the ethics reference number: HREC/13/SCHN/22.

**Funding:** This research did not receive any grant from funding agencies in the public, commercial, or non-for-profit sectors.

**Competing interests:** The authors have declared that no competing interests exist.

reliable, honest information and open-access visiting. Notably, fathers seek greater recognition of their role in the NICU—beyond being the *'other'* parent.

## Introduction

In Australia birth defects affect up to five percent of all infants and remain a leading cause of infant mortality [1,2]. Many birth defects are structural, requiring surgical intervention soon after birth. Outcomes and survival rates in surgical neonatal units have improved [3,4] both due to early intervention, and advancements in surgery and technology [5]. With more infants surviving newborn surgery, health-care professionals are recognising a greater need to focus on families as well as infants to provide better outcomes for the whole family [6].

Studies of parental needs in the Neonatal Intensive Care Unit (NICU) have predominantly focussed on mothers, particularly mothers of preterm infants [6,7]. In samples comprising both parents, mothers typically form the majority. Using the Neonatal Family Needs Inventory (NFNI), Ward [8] compared the needs of 10 fathers with 42 non-related mothers and found fathers ranked support, information and assurance needs significantly less important than mothers. In contrast, Mundy [9] found no significant differences between the needs of 43 mothers and 17 fathers. Although samples in these two studies comprised parents of preterm and term infants, parents of infants requiring neonatal surgery were not independently reported. Only one study has looked at fathers in a surgical NICU [10]; reporting that for 22 fathers stress was predominantly associated with alteration in parental role [10]. To date, there is limited information about fathers' needs in NICU, particularly fathers of infants requiring neonatal surgery [6] and quantitative studies.

Fathers of infants admitted to a NICU play an important role in supporting mothers and infants. Because mothers may be too unwell to accompany their infant, fathers are frequently the first point-of-contact between family and NICU personnel [11] and often the decision-making parent regarding any urgent treatment required. Fathers' family and social responsibilities as well as work commitments are widely recognised [11,12], however, the role of fathers in the NICU is less well-defined.

An emerging evidence-base from qualitative studies has revealed several themes in the experiences of fathers of premature infants and fathers' involvement in NICU. These relate to the need for quality information, maintaining a sense of control, participation in infant caregiving and decision-making, being treated as a unique individual, and the availability of 'father-specific' support [13,14].

This study aimed to identify the needs of fathers in a surgical NICU and determine whether their needs were being met by NICU health-care professionals.

## Materials and methods

Approval for the study was obtained from the Internal Ethics Review Committee of the Children's Hospital at Westmead (HREC/13/SCHN/22) prior to recruitment. Written informed consent was obtained from participants.

### Study design and setting

This prospective cohort study was conducted in a 23 bed, surgical NICU attached to a quaternary referral and teaching children's hospital in Sydney, Australia from January 2014 to September 2015. All babies are out-born and require transfer to a surgical NICU. Fathers received

an information sheet explaining the study purpose, primary investigator's contact information, and that choosing not to participate in the study would not affect care of their infant. Participants provided written informed consent. Sample size was based on previous annual admission numbers and the descriptive nature of the study. Data analysed for the current paper formed part of a larger study [15].

## Sample

Fathers of newborn infants admitted for surgical treatment of a congenital anomaly and present in the NICU between 48 and 72 hours of admission and literate in English were invited to participate. Fathers not literate in English (n = 4) were excluded because outcome measures were available in English only. A study of parents whose newborns exclusively required cardiac surgery was simultaneously in progress; due to participant burden we did not approach these fathers.

## Outcome measures

Fathers' needs were identified using the Neonatal Family Needs Inventory (NFNI) [8, 16]. This consists of 56 statements designed to measure the importance of needs across five sub-scales: Support (interpersonal and emotional support); Comfort (personal physical comfort); Information (communicating information about their infant and psychosocial support); Proximity (nearness to infant); and Assurance (feel confident about care given and outcome) (18; 7; 11; 8; 12 items, respectively). Participants rate each item statement as not important (1), slightly important (2), important (3), very important (4), or not applicable (5). This is the only tool available specifically for parents in NICU. It has high face validity and, at tool development, content validity was established using an expert panel and parents [8]. The NFNI showed good internal consistency with this sample (Cronbach alpha of 0.91), similar to that reported by Ward at tool development [8]. For the current study, fathers were also asked to indicate (yes or no) next to each statement whether NICU health-care professionals had met that need.

Fathers also completed the 13-item version of the Social Desirability Scale (SDS), [17] responding True/ False to statements that 'describe the sort of person you are'. This tool measures the tendency to answer questions in a manner viewed favourably by others. Eight items are reverse-scored; yielding a possible total of 13. High scores may indicate response-bias. In this study the SDS was used to assess social-desirability bias in the needs-met response data.

## Procedure

Fathers were given the NFNI and SDS paper-and-pencil questionnaires by the researchers between 48–72 hours of their infant's NICU admission and asked to return these to the primary researcher (P.G). Fathers provided demographic information. Discharge planning included asking fathers to complete and return a second NFNI before leaving the hospital. Where necessary, fathers were requested by phone to return the questionnaire by post.

## Statistical analysis

Likert-scale responses for NFNI need items and needs-met questions were coded; 'Not-applicable' was coded to '0'. There were no missing values on outcome variables. Descriptive statistics are reported for fathers and infant demographics, item and subscale level analyses. Frequency distributions and means using SPSS [18] were used to determine fathers' most important and least important needs. Admission and discharge data were compared using

paired data from 23 fathers. Effect sizes for subscales at admission were calculated as Cohen's d using formula for paired data comparisons to avoid over-estimation [19]. Conventionally, a d-value of 0.2 is described as small, 0.5 as medium, 0.8 as large and >1 as very large; however, meaningful interpretation of effect size is context specific [20]. Due to some skewed distributions and small subgroup numbers, parametric and non-parametric techniques were used, with similar results. Parametric results are reported to allow comparison with other literature. SDS scores were summed and mean total SDS scores correlated to total number of needs met and number met of the 10 most important needs, using Pearson's correlation r and Spearman's rho.

Approval for the study was obtained from the Internal Ethics Review Committee of the Children's Hospital at Westmead (HREC/13/SCHN/22) prior to recruitment.

## Results

Fifty-nine fathers met the inclusion criteria; 49 agreed to participate (83% participation rate). Forty-eight fathers completed the questionnaires at admission (48/49, 98% response rate); of these 23 completed questionnaires at discharge (23/48, 48%). As shown in Table 1, the sample comprised predominantly well–educated, employed, married fathers; the majority (85%) were less than 40 years of age; and for most this was their first child (28, 60%). No significant relationships were found between father demographics.

Most infants were term-born (33, 69%). Table 2 shows infant surgical diagnoses. Most infants had gastro-intestinal disorders (37, 79%).

No significant relationships were found between father and infant demographics, nor any demographics with outcome measures.

### Fathers' most and least important needs

Fathers' ten *most* important needs at admission and discharge are presented in Table 3. Identifying the order of the most important needs is particularly relevant for informing clinical practice. At admission, five of these most important needs related to Assurance. At discharge, the ten most important needs included five new items. The importance of receiving prior orientation to the NICU increased significantly and the need to visit anytime had decreased significantly. 'To have questions answered honestly' was consistently (admission and discharge) the most important need for fathers.

The ten *least* important needs at admission and discharge are presented in Table 4. At admission, five of the ten least important needs were related to Support. The importance of these ten needs increased at discharge, with the need for clergical visits and comfortable furniture increasing significantly. Classes about premature babies and feeling it is acceptable to cry became more important to fathers at discharge.

### Needs-met and Social Desirability Scores (SDS)

The ten most important needs at admission were met by the health-professionals more than 92% of the time. The most important need—to have questions answered honestly—was met for 98% of fathers. There were no significant correlations between mean Social Desirability Score (M = 8.4, SD 2.58) and total needs met (M = 47.8, SD 6.97; r = .12, p = .407), or the number of needs met of the ten most important needs (M = 9.7, SD .93; r = .15, p = .279).

**Table 1. Sample demographics of fathers and newborn infants (N = 48).**

| Characteristics | Frequency |
|---|---|
| *Father characteristics* | *n (%)* |
| *Age group (years)* | |
| 18–35 | 30 (62%) |
| 36–40 | 11 (23%) |
| > 40 | 7 (15%) |
| *English as first language* | |
| yes | 37 (77%) |
| no | 11 (23%) |
| *Birth country* | |
| Australia | 34 (71%) |
| South-east Asia | 5 (10%) |
| Other countries | 9 (19%) |
| *Marital status* | |
| married | 35 (73%) |
| defacto | 13 (27%) |
| *Education level* | |
| university | 20 (42%) |
| post-secondary | 21 (44%) |
| higher secondary | 1 (2%) |
| secondary | 6 (12%) |
| *Employment status* | |
| employed | 44 (92%) |
| not employed | 4 (8%) |
| *First child* | |
| Yes | 28 (60%) |
| no | 20 (40%) |
| *Previous NICU experience* | |
| yes | 2 (4%) |
| no | 46 (96%) |
| *Attended antenatal tour (n = 26)* | |
| yes | 9 (35%) |
| no | 17(65%) |
| *Infant characteristics (n = 48)* | |
| *Gender* | |
| male | 28 (58%) |
| female | 20 (42%) |
| *Gestational age (weeks)* | |
| 28–34 | 2 (4%) |
| > 34–37 | 13 (27%) |
| > 37 | 33 (69%) |
| *Birth weight (grams)* | |
| < 1500 | 1 (2%) |
| > 1501–2500 | 10 (21%) |
| > 2501 | 37 (77%) |
| *Antenatal diagnosis* | |
| yes | 26 (54%) |
| no | 22 (46%) |

(*Continued*)

**Table 1.** (Continued)

| Characteristics | Frequency |
|---|---|
| *Father characteristics* | *n (%)* |
| *Died before discharge* | |
| *Yes* | 2 (4%) |
| *no* | 46 (96%) |
| *Length of stay (days)* | |
| mean (SD) | 21.7 (12.25) |
| median (IQR) | 17.0 (19.00) |
| minimum–maximum[a] | 5–53 |

[a] Three outliers (>3 SD's) excluded (68, 104, 179 days)

## NFNI subscale scores on admission and discharge

Subscale level analysis showed that at admission fathers rated Assurance (M = 3.8, SD 0.266) needs highest in importance, followed by Proximity (M = 3.6, SD 0.35), Information (M = 3.5, SD 0.40), Support (M = 3.1, SD 0.51) and Comfort (M = 3.1, SD 0.62). At admission, differences in subscale mean scores were statistically significant (all p's < .001), except for Proximity versus Information (p = .162), and Support versus Comfort (p = .643). Assurance showed the highest effect sizes (moderate to large, see Fig 1). Fig 1 presents the order of subscales at admission (n = 48) and discharge (n = 23) showing only Support and Comfort changed place. At

**Table 2. Infant surgical diagnoses (N = 48).**

| Surgical diagnosis | Frequency n (%) |
|---|---|
| *Gastro-intestinal* | |
| Tracheo-oesophageal atresia/ fistula | 11(23%) |
| Gastroschisis | 6 (13%) |
| Duodenal atresia | 8 (17%) |
| Imperforate anus | 3 (6%) |
| Hirschprung's disease | 4 (8%) |
| Congenital malrotation | 3 (6%) |
| Cleft lip /palate with multiple anomalies | 1 (2.1%) |
| Exomphalos | 1 (2.1%) |
| Meconium ileus | 1 (2.1%) |
| *Total* | *37 (79.3%)* |
| *Respiratory* | |
| Diaphragmatic hernia | 4 (8%) |
| Congenital cystic adenomatoid malformation | 1 (2.1%) |
| *Total* | *5 (10.1%)* |
| *Genito-urinary* | |
| Congenital hydronephrosis with posterior- urethral valves | 3 (4%) |
| Bladder exstrophy | 1 (2.1%) |
| *Total* | *4 (6.1%)* |
| *Neurological* | |
| Spina bifida with myelomeningocele | 2 (4.5%) |
| *Total* | *2 (4.5%)* |
| *Total* | *48 (100%)* |

**Table 3. Ten most important needs of fathers at admission and discharge.**

| Ten most important needs at admission (N = 48) | NFNI Subscale | Mean score | SD |
|---|---|---|---|
| To have questions answered honestly | Assurance | 3.96 | 0.202 |
| To know NICU staff provide comfort to my infant | Comfort | 3.94 | 0.245 |
| To visit my infant anytime[a] | Proximity | 3.92 | 0.279 |
| To know the expected outcome | Assurance | 3.92 | 0.279 |
| To be assured best care provided | Assurance | 3.90 | 0.309 |
| To know my baby is treated for pain | Assurance | 3.90 | 0.371 |
| To know about medical treatment | Information | 3.90 | 0.309 |
| To know exactly what is done to my baby | Information | 3.88 | 0.334 |
| To know hospital staff care about my baby | Assurance | 3.88 | 0.334 |
| To see my baby frequently | Proximity | 3.88 | 0.334 |
| **Ten most important needs at discharge (N = 23)** | | | |
| To have questions answered honestly | Assurance | 4.00 | 0.000 |
| To know hospital staff care about my baby | Assurance | 3.96 | 0.209 |
| To know exactly what is done to my baby | Information | 3.96 | 0.209 |
| To be called at home | Information | 3.96 | 0.209 |
| To know NICU staff provide comfort to my infant | Comfort | 3.91 | 0.288 |
| To see my baby frequently | Proximity | 3.91 | 0.288 |
| To know specific facts concerning my infant's progress | Assurance | 3.91 | 0.288 |
| To be allowed to help with my infant's physical care | Information | 3.91 | 0.288 |
| To have explanations of the NICU environment before entering for the first time[b] | Support | 3.91 | 0.288 |
| To be recognised as important in my infant's recovery | Assurance | 3.91 | 0.417 |

NFNI = Neonatal Family Needs Inventory; items in italics were consistent over time

[a] significantly less important at discharge (M = 3.74, SD = .449, t = 2.47, 95% CI (mean difference) = .04 − .40, p = .02)

[b] significantly more important at discharge (M = 3.43, SD = .843, t = -2.31, 95% CI (mean difference) = -.49 − -.03, p = .03)

discharge, the importance of each subscale increased but paired analysis showed the increases were not statistically significant.

Error bars: +/- 1.4 x standard error of the mean (SEM). Effect sizes (Cohen's d) for subscales at admission based on comparisons using dependent data (n = 48) were A vs P = 0.6, A:I = 0.7, A:S = 1.4, A:C = 1.2; P:I = 0.2, P:S = 0.96, P:C = 0.8; I:S = 0.8, I:C = 0.7; S:C = 0.05 where A = Assurance, P = Proximity, I = Information, S = Support, C = Comfort. Comparisons using dependent t-tests (n = 23) showed no statistically significant differences on subscale scores between admission and discharge.

## Discussion

This study identifies the ten most and least important needs of fathers of newborns undergoing general surgery for major congenital anomalies. Identifying needs by order of importance informs evidence-based practice. The results demonstrate that fathers' needs may change between admission and discharge, and that needs were mostly met by NICU health-care professionals.

At both admission and discharge, fathers rated Assurance as most important, followed by Proximity and Information. This finding is similar to other quantitative studies that included fathers, however one of these studies looked only at very preterm infants [9] and the other

**Table 4. Ten least important needs of fathers at admission and discharge.**

| Ten least important needs at admission (N = 48) | NFNI Subscale | Mean score | SD |
|---|---|---|---|
| To have pastor /clergy to visit | Support | 2.08 | 1.217 |
| To have someone to help bring me to the hospital | Support | 2.33 | 1.191 |
| To have a phone near the waiting area | Comfort | 2.48 | 1.220 |
| To have comfortable furniture | Comfort | 2.71 | 1.031 |
| To have support groups | Support | 2.73 | 1.026 |
| To have classes on premature infants | Information | 2.79 | 1.237 |
| To have a bathroom near the waiting area | Comfort | 2.92 | 1.048 |
| To feel alright to cry | Information | 2.94 | 1.119 |
| To be shown concern about my health | Support | 2.96 | 1.220 |
| To help with the reactions of my infant's siblings | Support | 2.96 | 1.220 |
| **Ten least important needs at discharge (N = 23)** | | | |
| To have a phone near the waiting area | Comfort | 2.61 | 1.340 |
| To have someone to help bring me to the hospital | Support | 2.78 | 1.126 |
| To have pastor /clergy to visit[a] | Support | 3.13 | 1.100 |
| To be shown concern about my health | Support | 3.17 | 0.937 |
| To have a bathroom near the waiting area | Comfort | 3.23 | 1.066 |
| To have another person with them when visiting the NICU | Support | 3.30 | 0.822 |
| To have comfortable furniture[b] | Comfort | 3.43 | 0.843 |
| To have support groups | Support | 3.43 | 0.945 |
| To have reading materials about my infant's medical condition | Information | 3.43 | 0.896 |
| To help with the reactions of my infant's siblings | Support | 3.45 | 0.858 |

NFNI = Neonatal Family Needs Inventory

[a] significantly more important at discharge (t = - 3.54, 95% CI (mean difference) = - 1.38 − -.36, p = .002)

[b] significantly more important at discharge (t = -2.61, 95% CI (mean difference) = -1.09 − -.13, p = .016)

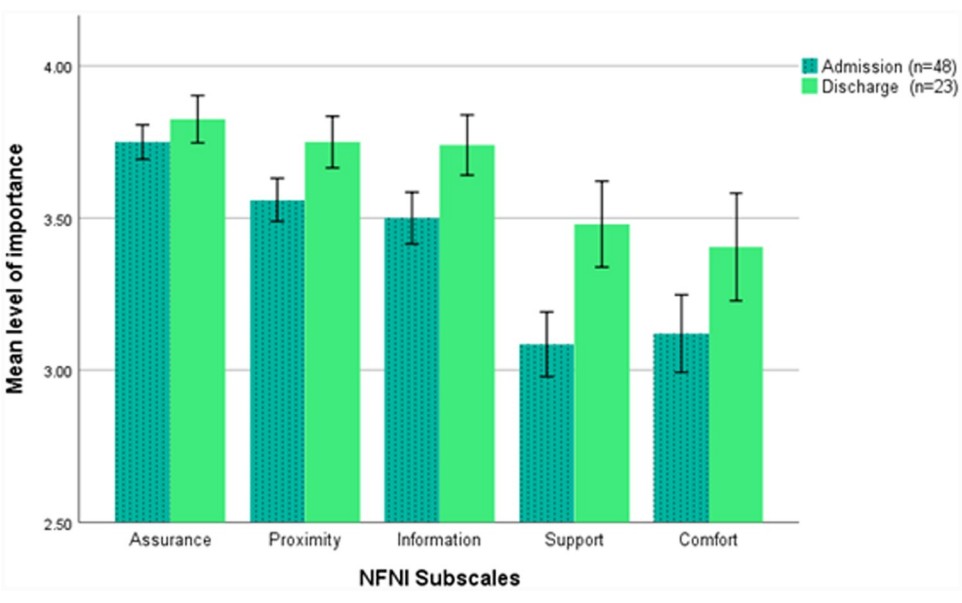

**Fig 1. Mean importance of neonatal family needs inventory subscales for fathers at admission and discharge.**

involved 10 fathers of infants without surgical conditions [8]. Our results highlight that assuring fathers warrants the attention of NICU health-care professionals in their clinical practice. At discharge Comfort gained higher priority than Support, but changes were not statistically significant. No comparative findings for this result have been reported in the literature.

Items on the Assurance subscale relate to information that parents find reassuring about infant's care and outcome. Five of the ten most important needs at admission relate to assuring fathers; in particular they want to be given honest information about prognosis, have questions answered honestly, know their baby's pain is well-managed, that their baby is getting best care, and that staff care about their infant. These needs align with the sense of security and control that are important for promoting fathers' involvement in the NICU [11,21].

Although most neonates in NICU undergo multiple painful procedures [22], fathers' focus on pain management may have been intensified because their infants had undergone painful surgical procedures and peri-operative care. It was also most important to fathers that staff attend to their infants' comfort. Such things as swaddling, containment and nesting, providing a pacifier and talking softly reassured fathers. Given that this NICU promotes individualised developmental-care [23], it is possible that fathers may have been influenced by staff prioritising these practices and, perhaps, witnessed benefits for their infants [24].

At admission, fathers' other most important needs related to Proximity (i.e. physical nearness and information promoting a sense of nearness to their infant), most importantly being able to visit anytime and frequently. At discharge, 'visiting anytime' was no longer among the ten most important needs, likely related to impending discharge or perhaps reflecting the open-access visiting policy of the study NICU. Seeing their infant frequently, however, remained among fathers' most important needs; reflecting the nearness that is important to the developing father-infant relationship [25].

The Information subscale relates to communication practices—specifically, conveying information and education, and communicating psychosocial support (e.g., it's alright to cry). At admission, fathers' most important needs included knowing about medical treatment and exactly what is being done for their infant. While keeping fathers informed remained important at discharge, knowing about medical treatment was replaced with wanting to know specific facts concerning their infant's progress and being called at home about changes in infant condition; possibly these relate to impending discharge. Fathers also placed greater importance on being shown how to help with their infant's physical care. These findings likely reflect fathers' change of focus to discharge and parenting at home. Fathers' need for information is consistently identified as a priority across studies [11,21,26].

Notably, there was only one item from the Support subscale (interpersonal emotional support) among fathers' ten most important needs. The need to have explanations of the NICU environment before entering for the first time became significantly *more* important at discharge than it was at admission. Perhaps initially more urgent matters take greater priority and some fathers may be dealing with shock [24]. This result accords with qualitative findings that highlight the need for strategies promoting fathers' sense of control through knowledge and information [11,21,26].

Interestingly, at discharge fathers placed greater importance on wanting to be 'recognised as having an important role in their infant's recovery' and 'being shown how to help with their infant's physical care' than they did on admission. The finding is concerning because it may suggest that fathers were not given adequate recognition and involvement in the NICU. Evidence from qualitative studies indicate that although fathers want staff to prioritise mothers, they also want to be seen as individuals with an important role beyond 'support' and want to establish a unique relationship with their infant [11,14,21,25,26,27]. These studies and our findings suggest that despite family-centered care practices it seems health-care professionals continue to focus on

mothers. NICU health-care providers are well-placed to offer greater assurance to fathers and to acknowledge their unique role in infant well-being. Over the past decade, the pivotal role fathers play in infant and child development has received wider attention [28], suggesting greater emphasis is warranted on supporting the role of fathers in the NICU.

Five of the *least* important needs on admission were from the Support subscale. Other studies that included fathers have reported similar findings [8, 9]. Although remaining among the least important needs, having comfortable furniture and a pastor or clergy visit were significantly more important at discharge. Perhaps when the infant is no longer gravely-ill, and fathers have constantly juggled commitments outside the NICU (e.g., work, sibling-care), they are more likely to identify practical needs relating to their own comfort and support [7,25]. Interestingly, our findings align with others that show fathers prefer to seek support from external sources (rather than support groups in the NICU) [11,21,27,29]. While similar findings have been reported [8,9] it is also possible that the term 'clergy' was not culturally-sensitive for multi-denominational Australia.

Items related to personal physical comfort and interpersonal/emotional support (including parent support groups) were consistently rated among fathers' lowest needs. This may reflect fathers' focus on their critically-ill infant and their tendency to prioritise the comfort and support needs of the mother and infant above their own. This finding is supported in a recent review by Ireland et al., [14] which concludes that most fathers generally prefer a 'background' supportive role and give priority to the needs of mothers and infants.

## Strengths and limitations

This appears to be the first reported study on the needs of fathers of infants undergoing general surgery in an NICU. As such it is difficult to assess the representativeness of the sample, and the generalisability of findings to the population of fathers in surgical NICUs. However, because the study comprised fathers who were predominantly highly-educated, employed, and married the sample may not be reflective of the population as a whole and their responses may have limited generalisability. Further, our sample included only fathers who were literate in English; this could be an area for other researchers to explore.

The sampling method may have been a possible limitation as only fathers present within 48–72 hours of the NICU admission were approached as this is the period during which surgery is most likely to happen. It is also when fathers may be 'juggling' commitments [14, 25]. That ten fathers (10/59, 17%) declined participation due to time-constraints suggests consideration is needed for fathers who face responsibilities outside of the NICU. The demographic results, however, suggest that infants in this study are broadly-representative of neonates in NICU's who undergo general surgery [3].

A strength of this study was the number of fathers who participated. This is considerably larger than previously reported in quantitative studies which included fathers whose infants required surgery [8–10]. Overall, studies of fathers' experiences in NICU (both qualitative [13] and quantitative) have predominantly focussed on premature and very-low birthweight infants, resulting in a paucity of evidence specifically about fathers of newborns requiring surgery for major non-cardiac congenital anomalies [6]. Despite the challenges of recruiting fathers in research, we achieved a recruitment rate of 83% (49/59) and 48% follow-up. Fathers were asked to complete and return questionnaires before leaving the hospital; those who did not were contacted by research personnel, with minimal response.

Other studies have excluded neonates with 'unknown prognosis', yet data from these fathers would likely enrich the evidence-base. The current study included two fathers whose neonates died before NICU discharge; an insufficient number for robust comparisons.

The study also explored whether the needs of fathers were met by NICU health-care professionals. To the authors' knowledge this has not previously been reported. Notably, fathers' ten most important needs were very well met. Further, SDS scores showed no evidence of social-desirability bias in fathers' responses.

This is the first Australian study we are aware of to use the NFNI, and more evidence is needed of its validity in this context. Further, the use of this tool has not been widely-reported. There are number of considerations regarding self-report measures. Even though fathers were advised that their responses were confidential, it may be that fathers are reluctant to admit the importance of their needs; implying perhaps higher levels of importance than our results showed. Our findings suggest that the NFNI may be appropriate for use with fathers in a NICU setting and may have validity to discern unique and changing needs.

## Conclusion and clinical implications

The need for assurance is a priority for fathers of neonates in a surgical NICU. Fathers are particularly concerned about pain management and infant comfort. Health-care professionals are relied upon to provide reliable, honest information and open-access visiting. A multi-layered approach to NICU practices that includes individualised family-centered care is recommended to best meet fathers' needs. Our findings suggest fathers want to be actively-involved and that fathers seek greater recognition of their role in the NICU—beyond being the *'other'* parent.

## Acknowledgments

We are grateful to all the fathers, Dr Peter Barr for his editorial comments and Mrs Claire Galea for her statistical assistance.

## Author Contributions

**Conceptualization:** Priya Govindaswamy, Kaye Spence, Nadia Badawi.

**Data curation:** Priya Govindaswamy, Sharon M. Laing.

**Formal analysis:** Priya Govindaswamy, Sharon M. Laing.

**Investigation:** Priya Govindaswamy.

**Methodology:** Priya Govindaswamy, Donna Waters, Kaye Spence, Nadia Badawi.

**Project administration:** Priya Govindaswamy.

**Resources:** Priya Govindaswamy, Nadia Badawi.

**Supervision:** Sharon M. Laing, Donna Waters, Karen Walker, Kaye Spence, Nadia Badawi.

**Validation:** Priya Govindaswamy.

**Writing – original draft:** Priya Govindaswamy, Sharon M. Laing.

**Writing – review & editing:** Priya Govindaswamy, Sharon M. Laing, Donna Waters, Karen Walker, Kaye Spence, Nadia Badawi.

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
