## [Decision Letter · Decision Letter 0]

5 Feb 2020

PONE-D-19-33546

Fathers’ needs in a surgical neonatal intensive care unit (NICU): Assuring the other parent

PLOS ONE

Dear Ms Govindaswamy,

Thank you for submitting your manuscript to PLOS ONE. After careful consideration, we feel that it has merit but does not fully meet PLOS ONE’s publication criteria as it currently stands. Therefore, we invite you to submit a revised version of the manuscript that addresses the points raised during the review process.

We would appreciate receiving your revised manuscript by Mar 21 2020 11:59PM. To enhance the reproducibility of your results, we recommend that if applicable you deposit your laboratory protocols in protocols.io, where a protocol can be assigned its own identifier (DOI) such that it can be cited independently in the future. For instructions see: http://journals.plos.org/plosone/s/submission-guidelines#loc-laboratory-protocols

We look forward to receiving your revised manuscript.

Kind regards,

Jayasree Nair, MBBS MD FAAP

Academic Editor

PLOS ONE

Journal Requirements:

1. Please include additional information regarding the scale or questionnaire used in the study and ensure that you have provided sufficient details that others could replicate the analyses. For instance, if you developed a questionnaire as part of this study and it is not under a copyright more restrictive than CC-BY, please include a copy, in both the original language and English, as Supporting Information.

2. Please provide additional details regarding participant consent. In the ethics statement in the online submission form, please ensure that you have specified whether consent was informed.

3.In your Methods section, please provide additional information about the participant recruitment method and the demographic details of your participants. Please ensure you have provided sufficient details to replicate the analyses such as: a) the recruitment date range (month and year), b) a description of how participants were recruited.

5. Please ensure that you include a title page within your main document. You should list all authors and all affiliations as per our author instructions and clearly indicate the corresponding author.

We appreciate that you have your title page upload separately, but please remove this file once you have included it within the manuscript file

Reviewers' comments:

Reviewer's Responses to Questions

**Comments to the Author**

1. Is the manuscript technically sound, and do the data support the conclusions?

Reviewer #1: Yes

Reviewer #2: Yes

2. Has the statistical analysis been performed appropriately and rigorously? 

Reviewer #1: Yes

Reviewer #2: Yes

3. Have the authors made all data underlying the findings in their manuscript fully available?

Reviewer #1: No

Reviewer #2: Yes

4. Is the manuscript presented in an intelligible fashion and written in standard English?

Reviewer #1: Yes

Reviewer #2: Yes

5. Review Comments to the Author

Reviewer #1: This is an excellent piece of work

1. Why did the authors choose the 48-72 hour time period after NICU admission for the 1st assessment?

2. Were the questionnaires anonymized or did the fathers know that the investigators would be able to identify them?

3. 23 fathers completed the questionnaire at discharge. What happened to the remaining 25? Did they refuse consent for the 2nd questionnaire? What were the demographic characteristics of the fathers who completed the questionnaire at discharge, and were there differences between them and those who did not answer the discharge questionnaire?

4. On page 6, line 122, it is not clear what is meant by “no significant relationships were found between father demographics”. Between father demographics and what?

5. Table 2 suggests that the surgical diagnoses were mutually exclusive. Were there no patients with more than 1 surgical problem?

6. Were the relationships between the responses at discharge and the duration of hospital stay, duration of NICU stay, type of surgical diagnosis, and outcome of surgical procedure, explored?

7. Does the unit have a policy of discharging all patients to home from the unit itself, or are there patients that are back-referred to a level II unit, and from there discharged home? Does “discharge” in this manuscript always mean being discharged home, or could it also mean discharge from the unit and being transferred to a stepdown care unit?

8. Did patients spend their entire time in the NICU, or were they transferred to a high dependency unit or ward before discharge? Was the set of doctors and nurses looking after the patient at the time of discharge the same as the one looking after the patient between 48-72 hrs after admission?

9. In the statistical analysis section, kindly clarify which paired tests were used for the responses on Likert scale?

10. For ease of analysis, it is probably okay to code “not applicable” as 0. But conceptually, it is not as though being “not applicable” is part of a continuum of responses that relate to the importance of a particular question. Coding it as 0 for the purpose of analysis implies that it occupies the least importance in the scale of importance, but that is not true. Had a “non-applicable” item been applicable for an individual subject, it is possible that it may have been accorded a great deal of importance. For example, availability of a clergyman may have been non-applicable for a large number of fathers, owing to their religious background; but had it been applicable, it may have been accorded importance.

Reviewer #2: This is a well-written manuscript describing the results of a prospective cohort study using surveys of fathers at admission and discharge to a surgical NICU. The methodology is sound, and results are presented in a format appropriate for a descriptive study. The change in fathers’ perceptions between admission and discharge has been presented well and relevant items are discussed. Validity of the tool used in this study was established at a previous study. The results of the study provides useful information about fathers' needs.

However, one major concern needs to be addressed. The authors mention that the data presented in this study formed part of a larger study. It is unclear if the entire paper is based on results using a subset of the data already presented on the previous paper by the authors’ group, or if any new surveys were added during the same timeline (reference 15- Govindaswamy P, Laing S, Waters D, Walker K, Spence K, Badawi N. Needs of parents in a surgical neonatal intensive care unit. J Paediatr Child Health. 2019;55(5):567–573. doi:10.1111/jpc.14249). It is important to clarify if this is, in essence, a subgroup analysis of results from their previous published study.

6. PLOS authors have the option to publish the peer review history of their article (what does this mean?). If published, this will include your full peer review and any attached files.

Reviewer #1: No

Reviewer #2: No

---

## [Author Response · Author response to Decision Letter 0]

25 Mar 2020

Manuscript Number: PONE-D-19-33546

Response to Academic Editor’s requests:

Dear Dr Nair, please see our responses to your instructions.

1. Please include additional information regarding the scale or questionnaire used in the study and ensure that you have provided sufficient details that others could replicate the analyses. For instance, if you developed a questionnaire as part of this study and it is not under a copyright more restrictive than CC-BY, please include a copy, in both the original language and English, as Supporting Information.

The two questionnaires used in the study are both published tools, the relevant references are given in the Methods section of the paper. The Neonatal Family Needs Inventory is copy-righted and was used in our study with permission from the tool developer. The Social Desirability Scale is freely-available (link now provided in the manuscript). We have amended the manuscript to better highlight the information that would assist others seeking to access these tools. 

2. Please provide additional details regarding participant consent. In the ethics statement in the online submission form, please ensure that you have specified whether consent was informed.

We have ensured that in the online submission form we have specified that consent was informed. As far as we could ascertain, we provided the required information, however, we have checked to ensure this information is specified.

3. In your Methods section, please provide additional information about the participant recruitment method and the demographic details of your participants. Please ensure you have provided sufficient details to replicate the analyses such as: a) the recruitment date range (month and year), b) a description of how participants were recruited.

We have added the following information to the Methods section of the paper: a) dates to the months and years that were stated in the paper; b) a description that more fully explains the process of recruitment, as shown below. 

Based on inclusion criteria, the primary investigator (PI) approached all eligible fathers and provided them with an information sheet explaining the study purpose, primary investigator’s contact information, and that choosing not to participate in the study would not affect care of their infant. Fathers who indicated to the PI that they wanted to participate were then asked to sign a written informed consent that included data collection at both admission and discharge, and given an envelope containing the anonymized questionnaires.

Thank you for drawing our attention to this. We have ensured that the titles appearing in these two places are identical.

5. Please ensure that you include a title page within your main document. You should list all authors and all affiliations as per our author instructions and clearly indicate the corresponding author. We appreciate that you have your title page upload separately, but please remove this file once you have included it within the manuscript file.

Thank you for clarifying this issue. We have removed the file as directed and included the title page in the main document.

Have the authors made all data underlying the findings in their manuscript fully available?

Reviewer #1: No

Reviewer #2: Yes

We note the discrepancies between the Reviewers responses’. Please note that we submitted the following declaration with the original manuscript as required: There are ethical and legal restrictions on sharing de-identified data set. Data from the study are available upon request, as there are legal restrictions on sharing these data publicly as these data contains sensitive and identifiable information. The data set contains details including birthweight, gestational age, gender, antenatal diagnosis, surgical diagnosis, fathers’ age, and birth country. Such information may be used to directly identify individuals, as the study site is one of only 3 state-wide referral centres in New South Wales for neonatal surgery of major congenital anomalies and each year there are relatively few neonates requiring surgery for each specific surgical diagnoses. The informed consent signed by the study participants and approved by the Internal Ethics Review Committee of the Children’s Hospital at Westmead, did not ask fathers about data sharing. Researchers who meet the criteria for access to confidential data may contact the corresponding author or the Executive officer, KIDS Research, Asra Gholami, 02-98453066 or asra.gholami@health.nsw.gov.au, and provide the ethics reference number: HREC/13/SCHN/22.

 

Manuscript Number: PONE-D-19-33546

Response to Reviewer #1.

Thank you for reviewing our manuscript and allowing us the opportunity to address your concerns. We have amended the manuscript to address your comments. Parts of the manuscript that related to your comments are now highlighted in yellow and new additions to the text appear in purple-coloured font.

Reviewer #1: This is an excellent piece of work

Thank you for your encouraging comment. 

1. Why did the authors choose the 48-72 hour time period after NICU admission for the 1st assessment?

Thank you for highlighting that this information needed to be more clearly stated in the manuscript. We noted this time-frame and the reason for choosing it in the Discussion section, but acting on your comment we have added the following information to the Sample section of the Methods: This time-frame was chosen as it is the period during which surgery is most likely to happen and fathers are most likely to be present. 

2. Were the questionnaires anonymized or did the fathers know that the investigators would be able to identify them?

Names of fathers were not collected on the questionnaires. Following consent, the primary investigator assigned participating fathers a study identification code based on the baby’s study identification number. These codes were noted on the questionnaires to allow matching of fathers’ responses at the two data collection time-points and to their infant data. Codes were known only to the primary investigator, kept confidential and stored in accordance with ethics requirements. This is now stated in the Methods section of the manuscript under the heading Procedure: Fathers who indicated to the PI that they wanted to participate were then asked to sign a written informed consent that included data collection at both admission and discharge, and given an envelope containing the anonymized questionnaires. Participating fathers were assigned a study identification code based on the baby’s study identification number. Codes were known only to the primary investigator, kept confidential and stored in accordance with ethics requirements.

3. 23 fathers completed the questionnaire at discharge. What happened to the remaining 25? Did they refuse consent for the 2nd questionnaire? What were the demographic characteristics of the fathers who completed the questionnaire at discharge, and were there differences between them and those who did not answer the discharge questionnaire?

Thank you for drawing our attention to clarifying this aspect. At the time of consent, fathers were informed that they would be asked to repeat the questionnaire at the time of discharge, and that they were consenting to participate in data collection at both admission and discharge. All 48 fathers’ received a second (repeat) NFNI questionnaire at least two days prior to planned-discharge. They were asked to either return the questionnaire before leaving the hospital (by handing the sealed envelope to a member of the research team or by placing it in the locked box located near the unit for this purpose) or use the enclosed return-addressed envelope to mail the survey to the primary investigator (PI). Twenty-five fathers did not return the questionnaire. The PI monitored the return of questionnaires and, using the study codes accessible to her only, phoned fathers whose questionnaires had not been received within the week following discharge. This, however, did not result in further questionnaires being returned. Please note that we have added these further details to the Procedure section of the Methods.

To address your other comment, we have added the following to the Results: No significant differences were found on any infant or father characteristics between the fathers who did (n=23) and those who did not (n=25) complete NFNI questionnaires at discharge. The small subgroup numbers, however, must be considered when interpreting these results. 

Given the lack of significant differences and consideration for space, we thought it unnecessary to report the demographics for these two groups of fathers separately. 

4. On page 6, line 122, it is not clear what is meant by “no significant relationships were found between father demographics”. Between father demographics and what?

Thank you for drawing our attention to this and please accept our apologies for this typographical error (and our proof-reading oversight!). This sentence now reads: No significant relationships were found within father demographics, or between father demographics and outcome measures at either admission or discharge. Interpretation of these results, however, warrants caution due to some small subgroup numbers. 

5. Table 2 suggests that the surgical diagnoses were mutually exclusive. Were there no patients with more than 1 surgical problem?

Six infants presented with more than one surgical condition. None of these infants, however, underwent surgery for more than one of these conditions while in the NICU. As such, Table 2 presents the mutually exclusive conditions for which babies underwent surgery while in the NICU. We have added this information to the Results section: Six infants presented with more than one surgical condition, but none underwent surgery for more than one of these conditions while in the study NICU. As such, the surgical conditions presented in Table 2 were mutually exclusive. 

6. Were the relationships between the responses at discharge and the duration of hospital stay, duration of NICU stay, type of surgical diagnosis, and outcome of surgical procedure, explored?

Thank you for highlighting the need to clarify this information. We explored relationships and associations between all infant characteristics (all variables shown in Table 1, as well as the surgical conditions) and fathers’ responses both at admission and discharge. With the exception of death while in the NICU, we did not collect details regarding outcome of surgical procedure. As shown in Table 1 and mentioned in the Discussion, two babies died while in the NICU – a number that did not support robust subgroup analyses. With the exception of two babies who were transferred to another ward in the hospital, the study babies remained in the study NICU for the duration of their hospital stay. We have added this information as a note to Table 1.

The purpose of our study was to inform care-practices in the surgical NICU, with particular focus on informing strategies that may assist fathers at the very stressful time early in their infant’s NICU admission and when preparing for discharge, which for infants in this unit is predominantly discharge to home. As only two babies went to other wards in the hospital, this was not a sufficient number for robust subgroup analyses of discharge data. 

The relevant findings are highlighted in the Results, and we have added the following paragraph to the Discussion: No significant relationships were found between father characteristics, infant characteristics and fathers’ needs at discharge. Interpretation of these results, however, warrants caution due to small subgroup numbers. 

7. Does the unit have a policy of discharging all patients to home from the unit itself, or are there patients that are back-referred to a level II unit, and from there discharged home? Does “discharge” in this manuscript always mean being discharged home, or could it also mean discharge from the unit and being transferred to a stepdown care unit?

The study unit is a level-3 NICU that provides level 6 care, which is defined by the New South Wales Department of Health as care that includes specialist surgical services (https://www1.health.nsw.gov.au/pds/ActivePDSDocuments/GL2016_018.pdf). The unit does not have a definite policy stipulating that infants can be only be discharged home. However, this tends to be the norm. Infants may be transferred to other wards in the hospital or to level-2 units elsewhere if extended time is required for establishing feeds, or educating parents about on-going care in preparation for discharge home. 

A small portion of our surgical NICU population comprises babies needing surgery for other reasons, including preterm infants. These babies are transferred to our unit from other (generally perinatal) NICU’s. When post-operatively stable and fit for transport, they are transferred back to the referring NICU. All the babies in the study sample were discharged home from the study unit except two babies who were transferred to another ward within the hospital. 

We have added to the Discussion: Although the unit does not have a policy stipulating that infants can only be discharged home, this tends to be the norm. Infants may be transferred to other wards in the hospital or to level-2 units elsewhere if extended time is required for establishing feeds, or educating parents about on-going care in preparation for discharge home. All the babies in the study sample were discharged home from the study unit except two babies who were transferred to another ward within the hospital. It is possible that fathers’ needs at discharge may differ based on outcomes of surgery and transfer of the infant rather than discharge home. Obtaining sufficient numbers and diversity for predictive studies regarding risk factors and subgroup comparisons in this unique parent population would likely require multisite research.

8. Did patients spend their entire time in the NICU, or were they transferred to a high dependency unit or ward before discharge? Was the set of doctors and nurses looking after the patient at the time of discharge the same as the one looking after the patient between 48-72 hrs after admission?

Thank you for drawing our attention to this aspect. All the study babies remained in the study unit for the duration of their admission. Consequently, the same staff of doctors and nurses looked after the patients for their entire stay. The probability of a particular staff member being allocated to care for a baby is the same at admission and discharge, given consideration for the usual factors of staff skill-mix and infant acuity at the time of allocation. 

9. In the statistical analysis section, kindly clarify which paired tests were used for the responses on Likert scale?

Likert-scale responses were analysed using paired samples t-tests and Wilcoxon Signed-Rank Tests. The following has been added to the Statistical analysis section of the manuscript: e.g., paired t-tests and Wilcoxon Signed-Rank tests were used for comparing NFNI admission and discharge data.

10. For ease of analysis, it is probably okay to code “not applicable” as 0. But conceptually, it is not as though being “not applicable” is part of a continuum of responses that relate to the importance of a particular question. Coding it as 0 for the purpose of analysis implies that it occupies the least importance in the scale of importance, but that is not true. Had a “non-applicable” item been applicable for an individual subject, it is possible that it may have been accorded a great deal of importance. For example, availability of a clergyman may have been non-applicable for a large number of fathers, owing to their religious background; but 

As this is a self-report tool, fathers have assigned importance to items that were (presumably) personally relevant them. As such, and as you have noted, not all items may have been relevant (i.e. applicable) to all fathers. It appears that fathers chose the option of responding ‘not applicable’, rather than ‘guessing’ how important a need may be if it applied to them. Re-coding ‘not applicable’ to ‘0’ was necessary because using a rating of ‘5’ as per the original version of the tool would distort interpretation of the results. Please note that we are not allocating a ‘global’ judgement regarding the importance of an item across all contexts. Our coding is intended to reflect that for a particular individual in our study sample a particular item held least importance because it did not apply to that individual. 

As acknowledged, our findings are based on the responses of our study sample. We have noted the characteristics of the fathers in our study and the possible limitations these impose on generalisation of our findings. We have also noted that it is difficult to assess the representativeness of our sample, given the scant data available regarding this particular type of NICU father population. More research is warranted in this important area, with more diverse samples of fathers, and more publications reporting the findings in the literature would assist in developing an evidence-base for clinical practice. 

 

Manuscript Number: PONE-D-19-33546

Response to Reviewer #2.

Reviewer #2: This is a well-written manuscript describing the results of a prospective cohort study using surveys of fathers at admission and discharge to a surgical NICU. The methodology is sound, and results are presented in a format appropriate for a descriptive study. The change in fathers’ perceptions between admission and discharge has been presented well and relevant items are discussed. Validity of the tool used in this study was established at a previous study. The results of the study provides useful information about fathers' needs.

However, one major concern needs to be addressed. The authors mention that the data presented in this study formed part of a larger study. It is unclear if the entire paper is based on results using a subset of the data already presented on the previous paper by the authors’ group, or if any new surveys were added during the same timeline (reference 15- Govindaswamy P, Laing S, Waters D, Walker K, Spence K, Badawi N. Needs of parents in a surgical neonatal intensive care unit. J Paediatr Child Health. 2019;55(5):567–573. doi:10.1111/jpc.14249). It is important to clarify if this is, in essence, a subgroup analysis of results from their previous published study.

Thank you for your encouraging words regarding the study and our manuscript. We note your concern and hope that we are able to address it satisfactorily. We would like to clarify that the fathers’ data used for analysis in this paper comprised a subset of data from the larger study, and no new surveys were added during the same timeline. However, we would also like to clarify that while the paper presents results from a subgroup analysis of the larger dataset, the current paper presents fathers’ results not previously reported. Notably: comparison of fathers’ item level data (individual needs) at admission and discharge; reporting of fathers’ needs-met results, Social Desirability Scale scores for fathers, and relationship between these variables for fathers; fathers’ ten most important and ten least important needs at admission; changes in order of subscale importance for fathers between admission and discharge; statistically significant differences between subscales at admission and discharge for fathers; effect sizes (Cohen’s d values) for subscales for fathers at admission. Further, the discussion provides a comprehensive consideration of the findings that focuses on meaningful interpretation within the context of better understanding fathers’ needs and their fathering role in the NICU. 

Despite society’s changing notions about fathering and increasing recognition of fathers’ important role in family and infant well-being, fathers continue to be an under-researched group among NICU parent populations. In particular, there are few quantitative studies reported about fathers, and very little is known about fathers whose babies require general surgery for major congenital anomalies in the newborn period. Please be assured that our manuscript presents a novel and unique contribution to an evidence-base that would inform practices for better supporting fathers in the NICU.

---

## [Editor Report · Decision Letter 1]

9 Apr 2020

Fathers’ needs in a surgical neonatal intensive care unit: Assuring the other parent

PONE-D-19-33546R1

Dear Dr. Govindaswamy,

We are pleased to inform you that your manuscript has been judged scientifically suitable for publication and will be formally accepted for publication once it complies with all outstanding technical requirements.

With kind regards,

Jayasree Nair, MBBS MD FAAP

Academic Editor

PLOS ONE
---

## [Editor Report · Acceptance letter]

13 Apr 2020

PONE-D-19-33546R1 

Fathers’ needs in a surgical neonatal intensive care unit: Assuring the *other* parent 

Dear Dr. Govindaswamy:

I am pleased to inform you that your manuscript has been deemed suitable for publication in PLOS ONE. Congratulations! Your manuscript is now with our production department. 

With kind regards,

on behalf of

Dr. Jayasree Nair 

Academic Editor

PLOS ONE